# Determination of Bile Acids in Canine Biological Samples: Diagnostic Significance

**DOI:** 10.3390/metabo14040178

**Published:** 2024-03-22

**Authors:** Krisztián Németh, Ágnes Sterczer, Dávid Sándor Kiss, Réka Katalin Lányi, Vivien Hemző, Kriszta Vámos, Tibor Bartha, Anna Buzás, Katalin Lányi

**Affiliations:** 1Department of Physiology and Biochemistry, University of Veterinary Medicine, István u. 2, H-1078 Budapest, Hungary; nemeth.krisztian@univet.hu (K.N.); kiss.david@univet.hu (D.S.K.); hemzo.vivien@student.univet.hu (V.H.); bartha.tibor@univet.hu (T.B.); 2Department of Internal Medicine, University of Veterinary Medicine, István u. 2, H-1078 Budapest, Hungary; vamos.kriszta@student.univet.hu; 3Faculty of Pharmacy, University of Szeged, Zrínyi u. 9, H-6720 Szeged, Hungary; lanyi.reka.katalin@o365.u-szeged.hu; 4Institute of Food Chain Science, University of Veterinary Medicine, István u. 2, H-1078 Budapest, Hungary; buzas.anna@univet.hu (A.B.); lanyi.katalin@univet.hu (K.L.)

**Keywords:** bile acids, enterohepatic circulation, canine, LC-MS/MS

## Abstract

The comprehensive examination of bile acids is of paramount importance across various fields of health sciences, influencing physiology, microbiology, internal medicine, and pharmacology. While enzymatic reaction-based photometric methods remain fundamental for total BA measurements, there is a burgeoning demand for more sophisticated techniques such as liquid chromatography–tandem mass spectrometry (LC-MS/MS) for comprehensive BA profiling. This evolution reflects a need for nuanced diagnostic assessments in clinical practice. In canines, a BA assessment involves considering factors, such as food composition, transit times, and breed-specific variations. Multiple matrices, including blood, feces, urine, liver tissue, and gallbladder bile, offer insights into BA profiles, yet interpretations remain complex, particularly in fecal analysis due to sampling challenges and breed-specific differences. Despite ongoing efforts, a consensus regarding optimal matrices and diagnostic thresholds remains elusive, highlighting the need for further research. Emphasizing the scarcity of systematic animal studies and underscoring the importance of ap-propriate sampling methodologies, our review advocates for targeted investigations into BA alterations in canine pathology, promising insights into pathomechanisms, early disease detection, and therapeutic avenues.

## 1. Introduction

Domesticated dogs (*Canis lupus familiaris*) were the first animals tamed by human beings around 20–40,000 years ago, according to the recent anthropological consensus. Canines were strict carnivores originally as their relatives living in the wilderness, wolves and dingoes, are still carnivores. Living with humans changed their access possibilities to feedstuffs; therefore, dogs turned into facultative carnivores slowly. For a long time in history, dogs were kept for their usefulness, such as protecting herds, assisting in hunting, and for other functions. Nowadays, dogs are companion animals possessing full family member status in many cases. Their health is brought to the attention of their owners—or human companions as many of the owners considers themselves—resulting in an increased interest in healthcare activities and prevention measures for protecting the well-being of their animal companions [1].

Bile acids (BAs) are steroids of amphipathic characteristics containing a 24-carbon atom core with a pentanoic acid substituent and playing important role in the metabolic processes of a series of living beings. They are significant factors in cholesterol homeostasis, lipid absorption, and maintaining the bile flow. Bile acids can be used as biomarkers and signaling compounds for diagnosing diseases, such as hepatic and intestinal diseases, malignant tumors within and outside the digestive system, and obesity. Consequently, changes in the concentrations of bile acids in blood plasma or serum, urine, or feces can provide important medical information for the prognosis, diagnosis, and follow-up of several diseases that involve bile acid metabolism [2].

While bile acid formation, circulation, metabolism, their roles, and their prospective diagnostic values have been studied intensively in human medicine, animal healthcare issues are underrepresented. In bile acid studies, animals are mainly used as “further species” to widen the biological scale of the studies or as model animals for studying closer human healthcare questions that would be difficult to study on human beings. Targeted studies on animals with the aim of clarifying animal healthcare issues are much less frequent and systematic.

The present review aims to explore healthy and clinical conditions, in which BA profile tests are applied to various species. Our objectives include proposing appropriate methodologies and matrices for the diagnosis of different diseases in canines while also confronting different methods and matrices utilized for the determination of BA profiles. We intend to assess the applicability of data obtained from various species including humans, mice, and rats for use in dogs as well as to explore how different dog breeds and body sizes may impact the composition of BA pools within dogs. We also seek to analyze the trustworthiness of available knowledge on BAs and BA profiles in dogs, as well as to investigate whether there is accurate information regarding the reported changes in healthy and ill animals concerning these characteristics. Finally, our aim is to summarize crucial elements from our review for the sake of emphasizing the significance of analyzing BAs with regard to the early detection of relevant disorders as well as to therapeutic purposes in companion animals.

## 2. Bile Acids in the Body: Physiological and Biochemical Aspects

### 2.1. Enterohepatic Circulation

In the last few decades, bile acids have been the focus of several studies due to their wide-ranging relations with numerous frequently studied fields of physiology, microbiology, internal medicine, and pharmacology. The extensive metabolic effects of these molecules with diverse characteristics are based on their continuous enterohepatic circulation (their specific route in the body) and also on their molecular structure.

Primary BAs are endogenously synthesized from cholesterol in the liver of the majority of mammals. Cholesterol is conjugated with glycine or taurine as the final step of a 17-step reaction sequence catalyzed by 16 enzymes. An ester, ether, or amide linkage on a side chain carboxyl group or to one of the ring hydroxyl groups increases the solubility of BAs. In humans, dogs, and cats, cholic acid (CA) and chenodeoxycholic acid (CDCA) are the major primary BAs. Immediately after synthetization, BAs are released into the bile in conjugated form, which is a secretion with a relatively complex composition and is concentrated and stored in the gallbladder. Not negligibly, bile salts make up the majority of the dry matter of the bile [3,4,5,6,7,8,9,10].

In the gallbladder, the bile consists of up to 15 distinct BAs, but the three main ones—72.8% taurocholic acid (TCA), 20.3% taurodeoxycholic acid (TDCA), and 6.2% taurochenodeoxycholic acid (TCDCA)—make up 99% of the overall pool [1].

When a food is consumed, the gallbladder is prompted by the entero-hormone cholecystokinin (CCK) to discharge bile into the duodenum via the bile duct. In this process, BAs assist in breaking down and absorbing lipids and fat-soluble vitamins [11,12,13,14].

Within the gut, these compounds, leastwise the fraction escaping absorption, undergo a multi-step molecular structural further transformation by the so-called Hylemon–Björkhem pathway of the resident gut microbiota (see Figure 1). Thus, BAs can be considered both substrates for microbial reactions and microbial nutritive factors, in this way having an impact on the survival possibilities of microorganisms. Since the description of this relationship, BAs have been recognized as being vital for maintaining an appropriate intestinal microbiota [7,15,16,17].

Microbial transformations involve four main pathways, i.e., deconjugation, dehydroxylation, oxidation, and epimerization. Furthermore, the microbiota has the ability for sulfate esterification and to reconjugate BAs, thereby augmenting the variety within BA pools [7,9].

Deconjugation (7 alpha-dehydroxylation) is the cornerstone of the aforementioned key processes (see Figure 1). 7α-dehydrogenase (7α-HSDH) is the primary enzyme responsible for the synthesis of secondary BAs. It is encoded by bile acid inducible (bai)-genes found in certain microorganisms present in the distal small and large intestines, particularly *Clostridium* species (e.g., *C*. or *Peptacetobacter hiranonis*, *C. scindens,* and *C. hylemonae*). When the biochemical reactions of BAs in the digestive tract are disturbed at certain points, the ratio of primary and secondary BAs shifts, causing changes in their toxicity index and hydrophobicity, as well as in the host metabolic function [4,7,8,9,15,17,18,19,20,21,22,23,24,25,26,27,28,29,30,31].

BAs are a sizable class of steroids containing a carboxyl group in the side chain. The structure of BAs is modified through the molecular relocation, removal, or entry of various functional groups during reactions; however, the number and position of hydroxyl groups on the sterol ring are the primary distinguishing characteristics [32].

**Figure 1 metabolites-14-00178-f001:**
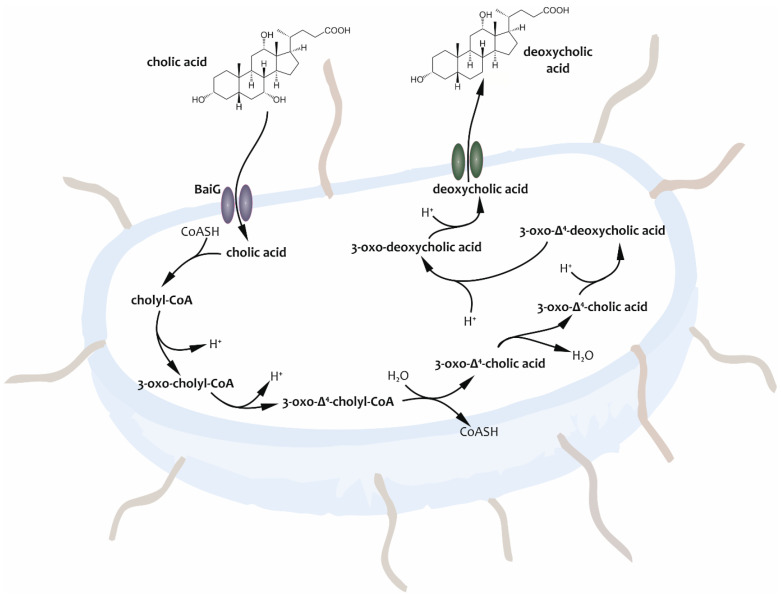
The process of cholic acid 7α-dehydroxylation, also known as the Hylemon–Björkhem pathway, occurs in intestinal anerobic microbes. Intermediate compounds formed in the synthesis of secondary bile acid deoxycholic acid are indicated [15,18,33,34].

During the enterohepatic circulation, the reabsorption of BAs is the step after microbial reactions. Some of their forms (deconjugated BAs) are absorbed passively from the jejunum to a small extent and by active transport from the terminal ileum and the colon in the main. The latter occurs via the apical bile salt transporter, or the apical sodium-dependent BA transporter (ASBT) expressed in the apical membrane of the enterocytes in the distal intestine epithelium, controlled by the so-called farnesoid X receptor (FXR) pathway, a nuclear receptor-binding mechanism of BAs. It may result in a potential defensive mechanism: the bile flow into the large intestine might be increased in response to an infection with invasive pathogens, which could modify FXR signaling and confer a degree of resistance to colonization. In the cytoplasm of enterocytes, BAs are capable of binding to the ileal BA-binding protein (IBABP) or the ileal lipid-binding protein (ILBP, fatty acid-binding protein, FABP) maintaining the transport and binding of BAs to FXR. Then, at the basolateral membrane, they are secreted into the blood by the heterodimeric organic solute transporter (OSTα/β) and return to the liver across the portal circulation. At this location, the cycle closes by getting BAs transported from the portal blood into the hepatocytes by the sodium taurocholate co-transporting polypeptide (NTCP) predominantly and by the organic anion-transporting polypeptides (OATPs) to a lesser extent. These polypeptides are located at the basolateral (sinusoidal) membrane, where they herewith activate the hepatic FXR system. FXR exerts an indirect negative feedback on de novo BA synthesis by regulating the expression of multiple genes implicated in primary BA synthesis. Then, initiating a new cycle, hepatocytes express an ATP-dependent efflux pump onto their membrane. This protein mediates the export of newly synthesized and recovered BAs to bile across the canalicular membrane (canalicular bile salt export pump, BSEP, ABCB11) [9,16,25,35,36,37,38,39,40,41].

A considerable proportion, precisely 95%, of the total BAs undergo reabsorption in the ileum before being returned to the liver for recycling and to modulate their de novo synthesis. The remaining 5% (the “loss”) is leaving the cycle eliminated via feces after being metabolized by the gastrointestinal microbiota in the colon. This ratio corresponds to the quantity of hepatic synthesis [3,9,11,21,39,42,43,44,45].

By impeding passive intestinal absorption, the conjugation facilitates the involvement of BAs in the intestinal digestion process of dietary lipids. Moreover, 10 to 50% of reabsorbed BAs enter the systemic circulation via peripheral spillover [11,36,44,46,47,48].

The enterohepatic circulation of BAs is shown in Figure 2, including cellular processes.

### 2.2. Cytotoxicity and Properties

As a substantial category of steroids, BAs have a carboxyl group attached to their side chains. During the preceding reactions, various functional groups are introduced, removed, or relocated molecularly to alter the structure of BAs. Nevertheless, the number and position of hydroxyl groups on the sterol ring serve as the principal differentiating features [50]. There are subsequent changes in their biochemical properties and effects on the host. However, it is relevant to mention critically that a contradiction can be found at this point from the side of different research groups, as well as in connection with the question of which fraction of BAs is toxic and which one is healthy.

In general, according to the most studies, secondary BAs are more hydrophilic than primary ones, and this fraction is meant to be the healthy one. Deoxycholic acid (DCA) and CDCA, which contain only two hydroxyl groups and are therefore less polar, are more hydrophobic, according to Zwicker and Agellon [20]. Biomembranes are typically more susceptible to cytotoxic effects as their hydrophobicity increases. Thus, as their cytotoxicity decreases, BAs are categorized as monohydroxy, dihydroxy, and trihydroxy, subsequently unconjugated, taurine-conjugated, and glycine-conjugated BAs [20,51]. Herstad et al. established a BA hydrophobicity scale that can be said basically as follows: ursodeoxycholic acid (UDCA) < cholic acid (CA) < CDCA < DCA < lithocolic acid (LCA) [52]. The significance of the hydroxyl group placement is further underscored by the protective nature of 7β-dihydroxy BA UDCA that is generated via bacterial transformation from the primary BA CDCA [20]. The secondary bile acids DCA and LCA are derived from CA and CDCA, respectively. UDCA is also produced when the primary bile acid CDCA undergoes bacterial transformation [1,52].

Nonetheless, Sitkin et al. conducted a study revealing that DCA could have either proinflammatory (bad) or anti-inflammatory (good) effects. Furthermore, their research team identified a Western diet as a risk factor for the onset of inflammatory bowel disease (IBD). In addition, the presence of elevated quantities of fecal DCA can have a crucial effect [53,54].

Concluding this chapter, one can establish that the classification of BAs into primary and secondary compounds pertains solely to their synthesis and does not necessarily indicate their positive or negative physiological effects. Questioning the classification of secondary BAs as either good or bad—as Lenci et al. stated in their review—is valid. Furthermore, the elimination of a hydroxyl group at the C-7 position leads generally to decreased water solubility and greater detergency compared to primary precursors. The correlation between the hydrophilic or hydrophobic nature of a particular BA and its potential for cytotoxicity and harmful consequences was revealed. From this standpoint, secondary BAs are commonly considered to be potentially harmful substances when they reach sufficient levels due to their detergent or destabilizing impact on cell membranes [55].

The most relevant canine BAs are shown in Table 1.

### 2.3. Roles

Apart from the fact that they are involved in digestive processes, called enteral lipid emulsification [46], depending on the state of the bile acid profile (BAP), BAs also have other local and systemic functions, or they can exert or amplify pathological effects.

On the one hand, these molecules exert influence within the enterohepatic circulation. While certain bile salts increase the effectiveness of antibiotics produced by bacterial species, others are capable of stabilizing the composition of the intestinal microbiome because they can be used as substrates for microbes [56].

In addition to the above, bactericidal activities of BAs on their own were recorded. In support of a previous hypothesis that the hydrophobicity of a BA molecule is associated with its antibacterial activity, Watanabe et al. documented their findings using the tenfold disparity in bactericidal activities between CA and DCA as an illustration. The amphiphilic nature of the molecule—the steroid skeleton of a BA molecule creates a hydrophilic side (α-face) and a hydrophobic side (β-face)—provides the context for this mechanism. It can lead to the disturbance of the integrity of bacterial cell membranes by causing a leakage of proton and potassium ions on them and consequently leading to cell death. [20,57,58,59]. Indeed, BAs are involved in the pathomechanism of some different liver diseases and inflammatory diseases of the gastrointestinal tract as well [3,9,10,30,44,54,56,57,60,61,62,63,64,65,66,67,68,69,70,71,72].

On the other hand, BAs were found to impact systems beyond the traditional scope of enterohepatic circulation. From some points of view, both the ab ovo synthesized and bacterially transformed BAs can be considered active metabolites with hormonal effects, the pathological patterns of which play a role in the manifestation of neuroinflammatory or neurodegenerative diseases mediated by the gut–brain axis [16,38,43,73,74,75]. These molecules can also lead to the improvement of barrier function and the mediation of anti-inflammatory mechanisms and can play a regulatory role at specific points of intermediate metabolism, e.g., in the metabolic regulatory system of carbohydrates or synthesis and oxidation of fatty acids [38,40,71,76,77,78].

Last but not least, the immunomodulating effect of BAs was also established [10,25,40,54,79]. As a brief insight into the mutual influence of BAs, the microbial environment of the gut, and the circumstances of the immune system, Ju et al. aimed to investigate the impact of gut BA levels on the gut microbiota and intestinal inflammation in subjects with IBD. The research team presented that the gut microbiota plays a crucial role in producing and modifying BAs, which can be disrupted by imbalances of, diversity in, and the quantity of gut microbes. BA metabolism issues affect the immune system and the function of intestinal barrier. The study found that alterations in gut microbiomes led to reduced levels of secondary BAs in serum and fecal samples, suggesting that the microbiome, BAs (and their metabolites), and IBD are correlated. DCA may cause alterations in the microbial community, which may have a significant role in the development of intestinal inflammation and is linked to disturbances in BA metabolism and the suppression of the FXR pathway. DCA also increased interleukin-1β concentration, decreased tuft cells, and enhanced cluster of differentiation (CD) 3+ and 4+ T cell expression in mice with drug-induced colitis, exacerbating intestinal inflammation. The regulation of intestinal BA levels is crucial for the clinical diagnosis and treatment of IBD [54].

Nevertheless, it is not to be forgotten either that their roles include not only those recorded recently but also trivial ones such as the regulation of cholesterol homeostasis. The process by which cholesterol is metabolized into BAs and bile alcohols is critical for its maintenance [32].

Over the last decade, the literature has been actively expanding with knowledge of the intestinal microbiota. Research data focus on the protective and pathological effects of the components of the microbiome on the gut–brain axis, as well as the influence of the bacterial composition with prebiotics and various exogenous substrates [9,26,29,63,73,75,80,81]. A complex parameter, the so-called dysbiosis index (DI), was developed to facilitate the progression in this research direction. While measuring the entire bacterial population quantitatively, this pioneering diagnostic parameter carries out also the quantitative characterization of seven specific bacterial groups separately (*Fecalibacterium* spp., *Turicibacter* spp., *Streptococcus* spp., *Escherichia coli*, *Blautia* spp., *Fusobacterium* spp., *Clostridium* or *Peptacetobacter hiranonis*) and provides information about the balance of the microbiome or its shift. The sensitivity of the procedure is low, and the specificity is high in terms of separating healthy dogs from patients [65,74,82,83].

## 3. Matrices to Be Analyzed for Bile Acid Composition

Since bile acids are synthesized in a well-defined area of the body and fulfil their role in another well-defined area, after which they are absorbed back into their synthesis place, the range of possible matrices for analysis is not too wide. Blood (serum or plasma), feces, urine, liver tissue, or gallbladder bile liquid are the most frequently studied matrices from this point of view. On the other hand, diagnostic purposes may require analyzing more than one matrix in order to gain a more complete view on the bile acid pool. Out of the almost a hundred bibliographic sources processed, only about half of them dealt with only one matrix, and approx. six percent had no matrix at all; they were articles focusing on basic method development working with pure solutions of bile acids. In all the remaining papers, more than one matrix was analyzed (see Figure 3).

Blood sampling could be an obvious choice, even in spite of the fact that the majority of bile acids are removed from the blood during enterohepatic circulation [84,85]. Considering blood samples, it is always a question whether blood plasma or blood serum is more suitable for analysis before deciding about more detailed issues. In the bibliographic sources studied, no significant difference could be found between bile acid concentrations in plasma and serum, with only a few exceptions; for example, Sarafian et al. reported lower bile acid concentrations in plasma [86]. According to the previous reviews in this field [87,88], there is a disagreement among authors concerning the suitability of blood serum or plasma as a matrix for bile acid analysis. Serum can be an argumentative decision since recoveries were found to be higher here than from plasma samples. On the other hand, the characteristics of plasma are more similar to those of whole blood due to the quenched coagulation [87]. This uncertainty could be tracked also in our study: 39.9% and 22.8% of the papers processed referred to serum and plasma, respectively, as the matrix studied; however, several times (9.6% of papers), serum and plasma were analyzed together. For 27.7% of the papers studied, blood was not used as a matrix for analysis. See Figure 4 and Table 2. The decision whether to use blood serum or plasma may depend on several factors, ranging from the technical background available on the sampling site to the aim of analysis in the laboratory.

Bile acids in blood serum or plasma samples can stay stable for 24 h at room temperature and for 2 months at −80 °C. Dried blood spots (DBSs), consisting of a small amount of capillary blood plasma or serum collected on a sampling card made of filter paper and dried afterward, can be stored even from 1 year to more than 18 years [87].

Total bile acid concentrations in human blood serum samples were found to be 2.1–8.0 μmol/L. CDCA, DCA, LCA, CA, and UDCA make up 92.1% of total BAs in the blood. CDCA, either conjugated or unconjugated, was found to be the most abundant BA. Blood plasma is similar to serum; they differ from each other mainly in containing the coagulation factor. Total bile acid concentrations in human blood plasma were found to be 1.4 to 6.5 μmol/L, so approximately the same order of magnitude as in the serum. The composition of bile acid species in plasma is similar to the serum, too: CA, UDCA, CDCA, DCA, LCA, and their glycine and taurine conjugates are the most abundant ones [87].

In human medicine, urine is a quite popular and obvious biological sample because it allows non-invasive sample collections which are easily available in most cases. On the other hand, urine can be a good choice as the matrix for mainly polar (hydrophilic) compounds that are excreted in themselves or in metabolized form through the kidneys. The analysis of highly hydrophobic compounds from urine could require special sample treatment and measurement methods [88]. In veterinary practice, urine is also an evident choice as a matrix for bioanalysis. Equipment for collecting urine samples on the spot or at a veterinary clinic are easily available and usable [87].

Bile acids may be less stable in urine samples than in blood serum or plasma. Specifically, longer standing at room temperature or repeated freeze–thaw procedures may cause substantial losses in BAs. Total bile acid concentrations in human urine samples were found to be approximately 9 μmol/L in a similar order of magnitude as in serum or plasma. DCA and CDCA were to be found the most abundant urinary BAs, comprising approximately half of the urinary bile pool [87].

Feces are solid- or semi-solid-state excretory products of digestion containing only a small portion of non-absorbed BAs. They also may be an attractive matrix choice due to similar reasons as urine (the possibility of non-invasive sampling). On the other hand, feces excretion is irregular; therefore, bile acids are not evenly distributed in any particular sample [87].

In the feces of adults, the BA pool consists of mainly unconjugated secondary BAs [87]. In contrary to a recent review in this field [88], we found that feces was the matrix analyzed in the second highest number in the bibliographic sources. Total bile acid concentrations in feces were found to be 1 to 25 μmol/g of dry matter. The majority of BAs in feces is present in unconjugated form and as secondary bile acids due to the bacterial activity in the guts. DCA and LCA were found to be the most abundant BAs in feces [87].

As we discussed before, bile is a biofluid produced by the liver, which is composed of mainly primary BAs and conjugated with glycine or taurine. Sampling from bile, duodenal or gastric fluid, or other liquids from the digestive tract requires an invasive intervention; therefore, it is applied only with specific precautions in both human and veterinary medicine and only when detailed information about the alterations of the BA pool is required [87,88].

Although bile is the primary medium for bile acids, their concentration in it is far from being even. The total BA concentration in bile may range between 0 and 180 mM, thus showing off a much higher variability than in blood, urine, or feces. The total BA concentration in gastric juice may range between 0 and 86 mM with usual average concentrations between 5 and 12 mM. The BAs of bile are mostly conjugated; however, the conjugated moiety depends on the subject’s age. While in fetuses, the majority of gallbladder BAs is conjugated with taurine, in the adults’ bile, most of the BAs are glycine-conjugated. The two primary bile acids produced by the liver, CA and CDCA, are found generally in bile samples [87].

In addition to the biological samples used most frequently for BA analysis, discussed above, other media can also be tested for them. Liver tissue was analyzed in 15 papers [28,63,89,90,91,92,93,94,95,96,97,98,99,100,101], although only 1 [99] included also dogs in the studied species. Sampling liver is especially difficult and requires highly invasive interaction with the subject sampled. Generally, it is performed by sacrificing the experimental animal, and therefore, mainly rodents are the test subjects.

Sometimes, ileal, caecal, duodenal, and other intestinal contents were analyzed for BAs. These studies involved mainly rats and mice, removing the intestinal content after their sacrifice [28,75,90,91,94], or humans were studied using a specific device to remove samples from the duodenal liquid [102]. Furthermore, as well, meconium [103], ovaries [101], saliva, exhausted breath [104], eWAT (epididymal white adipose tissue), muscle, and hypothalamus [95] were special matrices studied for BA content on a case-by-case basis.

**Table 2 metabolites-14-00178-t002:** Matrices studied the most frequently for BA composition.

Matrix		Bibliographic Sources
blood serum and plasma		[105,106,107,108]
	+ urine, feces, and bile	[109]
	+ urine	[86,110]
	+ feces	[111]
	+ liver tissue	[89]
blood serum and urine		[112,113,114]
	+ feces	[115]
	+ feces and bile	[116]
	+ feces and liver tissue	[90]
blood serum and feces		[54,64,67,117,118]
	+ bile	[75]
	+ liver tissue	[28,91]
blood serum		[119,120,121,122,123,124,125,126,127,128,129,130,131,132]
	+ bile	[133,134,135]
	+ liver tissue	[92,93]
blood plasma		[3,102,136,137,138,139,140,141]
	+ urine, feces and bile	[142]
	+ urine	[143]
	+ feces	[10]
	+ feces and liver tissue	[63,94]
	+ liver tissue	[95,96,97,98]
blood, without detailing whether it was serum or plasma		[144]
urine		[145,146,147,148]
feces		[19,52,65,83,149,150,151,152,153,154,155,156,157,158,159,160,161,162]
	+ bile	[103]
bile		[163,164]
liver tissue		[99,100,101]
no specific matrix (or pure solutions)		[18,57,104,165,166,167]

The sample preparation for BA measurements depends strongly on the choice of matrix and the range of compounds to be analyzed. The selection of analytical method may also influence the sample preparation procedure. Since all the biological samples used for BA analysis contain a rather significant amount of protein that may bind the bile acids and therefore modify the results of the analysis, deproteination is generally an important starting step usually made by adding organic solvents (methanol, acetonitrile), organic acids, or sometimes both of them together. Ultrasound homogenization or heating can also be applied together, possibly adding an alkaline agent to decrease BAs’ protein binding. However, this latter one should be carried out with care since bases may catalyze the hydrolysis of conjugated BAs [87,88]. The use of solid-phase extraction (SPE) is frequent to further clean and concentrate the samples; however, it is far from being a fully general method. From the papers we studied, only 24% used SPE for cleaning the biological samples. Evaporating the extract to dryness and reconstituting it into an organic solvent suitable for the chosen analytical method is the final step generally.

Recently, the extraction of bile acids from biological samples using various methods has been described several times. However, the efficiency of these extractions is not always comparable due to the amphipathic characteristics of bile acid molecules. This central question was addressed in a study aimed at comparing the critical efficiency for very low BA concentrations [116].

The irregularity of bile acid excretion makes the analysis of fecal bile acids more complicated and increases uncertainty of the measurements. Some authors suggested pooled feces aliquots of 4–5 days to be analyzed [87]. On the other hand, the liver and the intestines are very dynamic biological systems, and their status will fundamentally influence the bile pool, the composition of which changes dynamically with the organs producing and using it. The motility of the subject and the general healthcare status will also be an influencing factor. For all these reasons, pooling feces from 4–5 days may lead to losing valuable information about the details of the status and processes of the digestive system or the whole organism in general.

Because of its phase, composition, and origin, feces proves to be a more difficult sample than the liquid-phase biological samples. The preparation of feces samples generally involves the steps of lyophilization, homogenization, saponification, and extraction–preconcentration.

BAs are frequently measured from lyophilized feces instead of fresh ones [10,28,52,65,67,83,89,90,94,101,103,109,111,115,116,149,151,153,154,155,156,157,158], thus expressing the BA concentrations based on the dry weight of the lyophilizate. At the same time, it is important to mention that lyophilization requires special equipment and elongates the sample processing time. The interchangeability of fresh and lyophilized feces samples was studied [153], demonstrating that the measured BA concentrations were comparable for original and lyophilized samples. Original, fresh feces can be a suitable matrix for BAs analysis except for in detailed investigations of trace bile acids. On the other hand, the BA composition of feces does not refer to a given time point but instead a longer time period. Therefore, the comparability of feces BA concentrations to those measured in biological fluids is still to be clarified.

## 4. Measurement Methods and Instruments Used

Measuring the BA concentration can be conducted via two basically different approaches: quantifying the total bile acid amount present in the given sample or measuring the individual BAs one by one. Total bile acid measurement is an important part of present clinical laboratory practices; however, the determination of the individual BA concentrations has gained increased importance recently. Several reviews have dealt with the issue of bile acid analysis in the past few decades [2,32,84,85,87,88,168]. The common finding of these reviews is the statement that analyzing more bile acids together is a challenging task in the laboratory because of their low concentrations in biological samples (except bile fluid), similarity in chemical structure, and physico-chemical characteristics. The development of separation techniques and combining them with mass spectrometry has opened new possibilities in the analysis of BAs.

In addition to the two approaches described above, methods currently used for BA analysis can be divided further into two main groups. One of them includes those ones where the analysis is made without separating the bile acids from each other, and in the other one, the separation is carried out before the actual analysis. Analytical methods belonging to the first group are enzymatic assays, direct infusion mass spectrometry, or nuclear magnetic resonance (NMR) spectroscopy, among others [32,87].

Enzymatic methods are rather popular in clinical studies, owing to their relative cost-effectiveness and simplicity. Since many other enzymatic methods are also used in clinical laboratory practice, the general tools and expertise needed to carry out these tests are available in most cases. On the other hand, enzymatic tests may have specificity problems, and components of the matrix may cause cross-effects, spoiling the results. These methods have an even more significant drawback, i.e., analyzing several compounds needs one test per compound, or one can measure only the sum of the concentration of the compounds detected via the test. Therefore, enzymatic assays are expected to keep their significance in determining total bile acids (TBA) concentrations; however, they may lose their role in determining individual bile acids in the future.

Direct infusion mass spectrometry can be used for the preliminary screening of biological samples to settle the mass/charge ratios of the sought-after BAs. It is useful for obtaining quick qualitative information about the sample without requiring complex sample preparation; by contrast, it lacks the ability to allow a more detailed analysis. NMR spectroscopy is a powerful tool in the structure elucidation of organic compounds, which can give valuable information on the composition of biological samples. However, quantifications may be difficult, and the method requires expensive equipment and specialized expertise; therefore, its use for BAs analysis is not widely spread.

In the second group of bile acid analysis methods are those involving the separation of the compounds before or during the analysis. In the past few decades, the necessity of analyzing individual bile acids has become stronger, highlighting the importance of applying separation techniques (chromatography methods) in this field. From the many available separation procedures, several ones were tested for BA analysis. Thin-layer chromatography (TLC), supercritical fluid chromatography (SFC), and electrophoresis were all used, to some extent, for this purpose [2,168]; however, these ones gained no significant role due to their limitations in analytical performance, availability, or cost-effectiveness.

Gas chromatography (GC) was used to analyze bile acids in a few cases in the past [87]. Since BAs are not volatile, these procedures required derivatization that made sample preparation difficult and time-consuming. Traditional GC instruments did not possess any significant strength in bile acid analysis, even with the most carefully selected derivatization, therefore remaining in the background when analyzing BAs. This situation may change when the gas chromatograph is combined with a mass spectrometer (GC-MS). There is still need for derivatization to make BAs volatile; however, the powerful structure analyzing capabilities of electron ionization (EI) mass spectrometry used in GC-MS instruments may compensate for this weakness. It is still important to note that the derivatization process includes the cleavage of conjugates, therefore causing a significant information loss on the detailed composition of the bile pool studied [2]. Until the high-resolution mass spectrometry (HRMS) became wider spread in the LCMS methods, GC-MS was indeed a well-utilizable tool in BA analysis. Nowadays, its role is changing as GC-MS is losing its role in favor of more sophisticated LCMS methods.

High-performance liquid chromatography (HPLC) with optical detection methods can be a useful tool in analyzing bile acids since it copes well with non-volatile compounds. On the other hand, most BAs have weak absorption in the UV range (although there are exceptions); therefore, derivatization may be needed to improve the sensitivity [87,169]. Moreover, the separation of the structurally very similar bile acids, such as CDCA, DCA, and UDCA, can be challenging on traditional HPLC systems and columns. The introduction of ultra-high-performance liquid chromatography (UHPLC) with much narrower internal sizes in the equipment tubing, as well as smaller particle sizes and internal diameters in the LC columns, enabled the effective separation of the structurally similar BAs.

Combining UPLC systems with mass spectrometry (LC-MS), especially in tandem mass spectrometry systems, opened new horizons in bile acid analysis, and this is expected to continue [2,32,84,85,87,88,168,169]. Electrospray ionization is used in most cases in negative polarity, detecting [M-H]- precursors. The so-called multiple reaction monitoring (MRM) or selected reaction monitoring (SRM) are used most frequently for quantifying bile acids. The approach uses one or more mass spectrometric transitions from the same precursor, thus enabling better sensitivity and selectivity of the measurement method. Conjugated BAs can be conveniently sensed in that way since glycine and taurine units give stable and significant fragments of *m*/*z* 74 and 80, respectively. Unconjugated bile acids lack the stable [M-H]- precursors; therefore, these ones are frequently measured using the selected ion monitoring (SIM) approach when only the precursor ion is detected, or using the so-called pseudo-MRM when the [molar mass-1 → molar mass-1] theoretical transition is followed. However, for some unconjugated bile acids, more complex adducts can be detected in positive polarity ESI [166]. Still, it can be said that the fragmentation of bile acids is rather weak among ESI circumstances; moreover, the same fragments may appear for several compounds containing the same specific chemical moieties. Therefore, effective chromatographic separation remains of high importance when analyzing bile acids. In the case of the chromatographic separation of BAs for LC-MS measurements, chiefly, C18 columns are used. Solvents used as eluents include acetonitrile and, to a lesser extent methanol, with or without adding formic acid or ammonium formate. Aqueous solutions of formic acid, ammonium formate, or other organic buffers are most frequently used as the aqueous phase.

Mass analyzers used in the instruments performing BA analysis include triple quadruple (QqQ) MS-MS, time-of-flight (TOF), and ion trap (IT) units, the former one frequently connected to a quadruple analyzer (Q-TOF). From the bibliographic sources processed, approximately three-quarters of the papers used LCMS for measuring BAs, QqQ and TOF or IT MS sharing an approximately 2:1 ratio within it (see Figure 5). Single quadruple analyzers are seldom used since the identical molecular masses, the very similar structures, and the co-elution originating from these, in the case of some bile acid compounds, render single quad MS instruments less useful in BA analysis.

In addition to the targeted analysis of chemical compounds expected to appear in the sample, the semi-targeted or completely non-targeted analysis of additional related compounds, metabolites, or other transformation products has gained increased importance recently when analyzing bile acids in biological systems. While targeted analysis can be considered a routine process, the untargeted approach requires specific pre-processing methods and sophisticated statistical calculations to investigate the metabolite profiles of BAs [78].

## 5. Factors Influencing the Enterohepatic Circulation of BA

### 5.1. Effects of Feeding State on the Bile Acid Levels

The circulating total bile acid level is measured typically 8 to 12 h after feed deprivation and 2 h after consuming a small portion of canned feedstuff to stimulate gallbladder contraction. Fasting serum TBA in dogs is within the range 0 to 15 µmol/L, and the postprandial (PP) values range between 0 and 25 µmol/L, with a slight difference in cut-off values per laboratory [46,170]. During food intake, the degree of luminal tension to gastric wall elevates, and cholecystokinin (CCK) mRNS is upregulated in duodenal mucosal enterocytes, which is even more enhanced by the high lipid content of the bolus. Consequently, hormone synthesis increases, thus elevating the CCK level in the bloodstream. As a result of this process, the so-called choleretic effect succeeds to be manifested through an endocrine way, via contracting the gallbladder and relaxing the Oddi sphincter, which leads to the emptying of the gallbladder. The occurrence of this phenomenon can be prolonged, resulting in the formation of the highest concentration of PP serum BAs after more than 2 h following the release of CCK [46,171].

However, in healthy dogs, the gallbladder is shown to become empty in an even shorter period of time via the application of a CCK analogue rather than a simple test meal. Bridger et al. compared the effect of feeding-induced CCK provocation with CCK-analogue ceruletide stimulation on TBA levels. They found that the ceruletide-induced BA stimulation had been more sensitive than feeding-induced CCK. The direct ceruletide administration produces a more potent contraction of the gallbladder compared to contraction caused by a meal. The fluctuations in the rate, at which the stomach empties after eating, influenced by both individual factors and medical conditions, can result in the highest levels of BA in the blood occurring 1 to 8 h after consuming feed. Evidently, ceruletide serum BA stimulation is a viable and dependable method for diagnosing congenital PSS. The procedure is capable of being completed in 30 min as opposed to 120 min, and it is unaffected by the patient’s meal intake or vomiting episodes [13,172,173,174].

As further effects of elevated CCK levels, the gastric motility and emptying are inhibited by constricting the pylorus and relaxing the stomach, while oppositely, motoric movements of the colon are stimulated. Furthermore, CCK has a slight stimulating effect on the bicarbonate secretion, supporting the impacts of the secretin hormone and also leading to an enzyme-rich pancreatic juice secretion, provided that the carbon chain of the fatty acids in chymus consists of more than 8–12 atoms [13]. Additionally, through a negative feedback mechanism of the two hormones in question, a duodenal peristaltic and secretory inhibition is performed as well [14,173,174].

An irregular contraction of the gallbladder during fasting can lead to a preprandial level exceeding the PP value. Clinically, it is common practice to measure fasting values and PP bile acid values in pairs to acquire two comparable values in the dynamically changeable factor in question (Pena-Ramos et al., 2021). For preventing lipemia, a minimal quantity of the meal is administered to induce a response, and blood collection is conducted 2 h following the meal [46,173].

As the liver is responsible for keeping TBA in the physiological range, even postprandially, an insufficient function of this organ can be suspected in case of a too high fasting level and/or PP TBA level [46]. It is not without reason that the determination of paired TBA, which is usually called a PP stimulation test, is the most relevant liver function parameter. Resting serum bile acids exhibit greater specificity, but poorer sensitivity, than PP BAs. Measuring PP serum BAs can serve as a more effective screening test for excluding liver disease compared to measuring resting serum BAs [171]. Prolonged fasting, unsuitable test meals, compromised intestinal absorption, and increased food transit in the intestinal segments, together or respectively, may decrease BA levels and impact on the accuracy of bile acid testing for hepatobiliary diseases. BA tests should not be performed in animals showing clinical jaundice or with elevated levels of direct (conjugated) bilirubin. This test does not provide any data on liver function or the existence of portosystemic shunting in the presence of cholestasis.

In the bibliographic sources, there are numerous studies in which extremely elevated PP (and even fasting) BA concentrations were measured, especially in the case of CPSS and cirrhosis [171,175,176,177].

### 5.2. Impact of Motility and Transit Times

A number of factors influence the enterohepatic circulation of BAs, which consequently affect the TBA level. The transit time (TT) of the intestines, which is determined by their motility, is one of these factors. Evidently, variations in digestive anatomy can significantly affect physicochemical parameters including pH, digestive secretions, TT, and gut microbiota [1,14].

Due to their more developed colon and caecum, large breed dogs produce feces which is softer and moister than those of small breed dogs, according to an analysis by Weber et al. based on four PhD theses published between 1998 and 2013. Large dogs’ digestive mucosa exhibits increased permeability, which may cause an electrolyte backflow and account for the decreased consistency of their feces. This may account for the increased colonic fermentative activity and prolonged colonic TT. In addition, there was no significant correlation between the weight or size of the dog and the gastric emptying parameters nor a significant impact of the body size on small intestine TT, according to the review [48,178,179].

Gastric TTs, small intestinal transit periods, and maximal pressures during GI transit exhibit significant variations [180].

Having compared the gastrointestinal physiologies of cats and dogs, Tolbert et al. found that canines have shorter total TTs and quicker TTs. The increased potential capacity of the esophageal lumen in Beagle dogs and the reduced esophageal motility in cats account for this. In contrast to cats, dogs had a quicker gastric passage (with pre- and postprandial ranges of 6.8–15 h [1] and 31–829 min [181]) and a slower intestinal transit period (795–830 min). A reduction in gastrointestinal pH was observed in all species after feeding, with a substantially greater decrease in the first-hour small intestinal pH in dogs postprandially compared to cats [181]. The mean TTs in Beagle dogs pre- and postprandially were as follows: esophageal 3 min and 13 min; gastric 31 min and 829 min; and intestinal 795 and 830 min [181]. No correlation seems to occur between body weight (BW) and gastric emptying time, both in fed and fasted subjects. However, it is apparent that the consistency of feed has an impact on gastric emptying time. In addition, there is no consensus regarding the relationship between the dog size and the small and large intestinal TT. However, the available data indicate that the former is significantly shorter in the largest dog breeds. On the other hand, the data indicate a significant positive correlation between total TT and BW, highlighting the impact of breed as well as body size on this factor. The range is 22.9 to 31 h, 19.1 to 55 h, and 18.2 to 45 h for small dogs, medium dogs, and large dogs, respectively [1,181].

The pre-treatments had only a minor impact on the brief (0.57 ± 0.37 h) stomach TT in the fasting state. Gastric transit was evidently prolonged in the fed state (2.94 ± 0.91 h); compared to meat, a liquid meal resulted in a quicker gastric emptying time, with 90% emptying in 0.4 h and 50% in 1–3 h [1]. The study arms did not differ significantly in the maximum pressures detected in the canine GI tract; in the stomach, maximum pressures of up to 800 mbar were commonly reported [180].

The effects of biliary and pancreatic fluid diversion on canine upper gastrointestinal motility and hormone secretion were investigated by Sato et al. Upper gastrointestinal motility shows two types of patterns, being classified as either interdigestive or PP. The former pattern consists of periodically changing movements, the so-called migrating motor complex (MMC), while in the latter state, persistent contractions are performed. It is thought that BA pancreatic secretions regulate MMC. Although the MMC seems to decrease in canines undergoing total external biliary diversion, it reappears following BA administration. The research examined primarily the correlation between the MMC and bile or pancreatic juice; however, it is probable that the diversion of biliary or pancreatic juice modifies the properties of the MMC. Uncertainty surrounds the function of bile and pancreatic fluid in upper gastrointestinal motility after a meal [182].

Postprandial contractions in the upper gut were repressed. The duration of PP contractions in the proximal jejunum, which is correlated with gastric emptying, was prolonged. Additionally, the frequency and migration velocity of the MMC in the jejunum were reduced [182].

### 5.3. Impact of Breeds and Body Sizes

Due to decades of selective breeding, dog sizes have varied, which has been linked to differences in digestive physiology and susceptibility to disease. According to a review paper, dog sizes clearly show changes in interior anatomy. In dogs weighing 60 kg and 5 kg, the mature digestive tract length can account for 2.8% and 7% of the total body mass, respectively [178]. Colonic measurements, TT permeability, fiber degradation, fecal SCFA concentration, and fecal water content all rise with body size, but fecal bile acid concentration falls. Changes in microbiota composition based on dog BW may also be linked to modifications in BA concentrations, inversely correlated with BW, as certain bacteria are fully involved in BA deconjugation [178].

These findings imply that there are differences in microbiota activity between small and large breed sizes, particularly in the recycling of BA. Physicochemical characteristics including pH, TT, and digestive secretions can all be affected by variations in digestive anatomy, which in turn can have an impact on the microbes in the gut [1,178].

Canine body sizes also affect the fecal BAPs. As BW increases, there appears to be a decrease in both the primary to secondary BA ratios and TBAs. Deschamps et al. summarized the fecal BA concentration ranges of a few studies, according to which the values are most frequently between 5.1 and 7.5 μg of total BA per mg of dry feces. Data related to the ratio of primary and secondary BA content in feces were also collected. They indicate that fecal secondary BA (84.9%) has greater relative percentages than primary BA (15.5%). Furthermore, proportions of main BAs, such as CA and CDCA, appear to be connected with canine BW negatively, although secondary BA shows the opposite relationship (one study only). These findings imply that there are differences in microbial activity between small and large breed sizes, particularly in recycling BAs [1].

The former team created the CANIne Mucosal ARtificial COLon (CANIM-ARCOL), a new model of the canine colon, in order to overcome this limitation. This model replicates the key nutritional (ileal effluent composition), physiochemical (pH, TT, anaerobiosis), and microbial (lumen and mucus-associated microbiota) features of this ecosystem, and it is customized for three dog sizes: small (below 10 kg), medium (10 to 30 kg), and large (above 30 kg). In the application of this model, it was disclosed that body size had an obvious impact on daily BA concentrations, with small bioreactors exhibiting concentrations of about 200 µg/mL in the fermentation medium compared to 400 and 500 µg/mL in medium and large conditions, accordingly. The corresponding profiles showed a substantial difference also in relation to dog size, with a considerable rise in DCA and a drop in the LCA proportion in large dogs as opposed to small and medium dogs. Furthermore, the percentages of CDCA and a structural isomer of LCA tended to be larger and lower, respectively, in large bioreactors. Finally, there was a substantial rise in DCA and LCA concentrations with dog size as well as the luminal compartment’s ammonia concentration [48].

Uniquely, it was observed in a recent study that TBA concentrations increased in vitro with dog size. However, contrasting trends were noticed in dog stools. Deschamps et al. were the first who described the efficient deconjugation of primary BAs into secondary BAs in an in vitro canine model using a nutritive medium. Furthermore, the proportions of secondary BAs increased proportionally with dog size, which aligns with the results obtained from fecal samples collected in living organisms [48].

It should also be taken into account that the focus of hepatic BA conjugation in dogs differs from that in other commonly investigated species, leading to further fundamental canine specificities in BA pools. Kook et al. noted that in accordance with the previously published results for six dogs, taurine-conjugated BAs comprised more than 99 percent of the total BA pool. Their own study confirmed these results using the gallbladder BAP for all 12 dogs examined (i.e., 99.04% for taurine-conjugated BAs, 0.15% for glycine-conjugated BAs, and 0.81% for unconjugated BAs) [50]. While rodents are capable of forming both taurine and glycine conjugates, dogs exhibit a predilection for taurine conjugation, while non-human primates and humans show a preference for glycine conjugation [50,59].

A small percentage of BAs can be converted by dogs via glycine conjugation, whereas cats appear to endure obligate conjugation with taurine. It appears that even taurine-depleted cats generate negligible quantities of glycine-conjugated BAs [46].

### 5.4. Impact of Age and Aging

Lee et al. described a method for profiling BAs in bile. Age-dependent alterations in rat BAs were characterized by applying a method for profiling BAs in bile. Unconjugated BAs and glycine-conjugated BAs were observed to decrease or remained unaffected; in turn, taurine-conjugated BAs exhibited a general increase. Consistent with the patterns observed in the targeted BAs, five of the taurine-conjugated BAs showed an increase, whereas a glycine-conjugated BA demonstrated a decline among the unknown BAs. The concentrations of 19 targeted BAs in rat bile obtained from subjects aged six weeks, six months, and fifteen months were investigated in this study. The predominant forms of BAs were taurine-conjugated, which constituted more than 80% of the entire pool of BAs. The levels of unconjugated BAs and glycine-conjugated secondary BAs decreased as the subjects aged. The levels of taurine-conjugated primary BAs increased. The research revealed that the concentration of primary taurine-conjugated BAs had increased in the absence of bacteroid activity; on the other hand, the concentration of glycine-conjugated primary BAs had been unaffected. Additionally, the research defined 22 unidentified BAs in rat bile, which may serve crucial functions and require additional investigations [163].

Secondary BAs are known to hinder the growth and spore germination of *C. difficile*. A recent study demonstrated that this hindrance was influenced by the dosage of secondary BAs. Blake et al. found that the levels of secondary BAs in the feces of puppies up to 5 to 6 weeks were much lower compared to adult dogs. A simultaneous rise in the levels of secondary BAs in the feces and a decline in the abundance of *C. difficile* provided evidence to support the concept of correlation in dogs [19].

### 5.5. Impact of Nutrient Content and Microbial Aspects

A study on C57BL/6J mice found that obesity and weight loss induced by diet had altered BA concentrations and BA-sensitive gene expression in insulin target tissues. La Farano et al. postulated that certain alterations in BA concentrations and associated gene expression would persist following weight loss in diet-induced obesity. They found that obesity induced by diet decreased hepatic BAs while increased numerous conjugated BAs in the plasma. The investigation on BAs in various tissues revealed that the levels of BAs in the hypothalamus, muscle, and epididymal white adipose tissue had been variable but largely stable. The observed changes in BA levels were substantially reversed, irrespective of tissue, when the feed was changed from high fat to low fat, and the subjects returned to a leaner physique and an insulin-sensitive state. The team established that further research is needed to examine the effects of BA deposition in metabolically significant tissues in relation to weight gain and loss [95].

According to a study on healthy canines, an excess of dietary lipids can impair gallbladder motility and BA metabolism [46]. For the duration of the two-week study, seven healthy Beagles were administered either a low-fat diet or a high-fat and high-cholesterol diet. The former increased total cholesterol and TCDCA in the plasma, while it decreased CCK-induced gallbladder motility. In contrast, the latter decreased TDCA in bile. Comparable alterations in the composition of BA and biliary hypomotility can be observed in canines with hyperlipidemia, according to the findings [12]. Kakimoto et al. referred to studies applying muscle strips or smooth muscle cells from the gallbladder to illustrate how hydrophobic BAs inhibit smooth muscle contraction through the induction of lipid peroxidation and oxidative stress (high H_2_O_2_ concentrations), in addition to their potent detergent effect on tissues [12].

It may be beneficial to consider an additional aspect in future perspectives, i.e., the associations between the intestinal microbiome of dogs fed with various feed forms and the BAP. Our concept is backed by the established strict correlation between gut microbial composition and BAP, as well as the observation by Schmidt et al. The research group found that the metabolome and microbial communities of BARF-fed dogs were significantly distinct from those of commercially fed ones [158].

Given that both BA compounds and microbes are influenced by the feed consumed, which in turn causes substantial changes in the diversity of the gut microbiota, it is indisputable that the quality of feed applied also impacts BAP [183,184].

Inconsistent serum BA results can be caused by variances in the type or amount of feed applied to the PP serum BA stimulation test, as well as differences in stomach emptying and gallbladder contraction [13].

To emphasize the relevant connection between the nutrient content of a feed and the state and species composition of the gut microbiota, Belchik et al. used a type of prebiotic to investigate the microbiome and the BAP in antibiotic-treated dogs. Supplementing with novel milk oligosaccharide biosimilars helped the dogs receiving antibiotics to recover from antibiotic treatment more quickly, to preserve microbial diversity, and to have more consistent fecal scores. The gut microbes and secondary BAPs are crucial for gut health. They revealed that dogs employ mainly *C. hiranonis* as a BA converter, whereas humans require primarily *C. scindens* in BA dehydroxylation reactions. It is known that in sick dogs, the chance of relapse is higher when secondary BA profiles and *C. hiranonis* cannot be recovered [184].

To conclude, under physiological conditions, the dynamics of bile acid circulation in canines is modulated by factors, such as the composition of the feedstuff, transit time, motility patterns, and the intestinal microbiome. Beyond these, the BA pool exhibits notable diversity with respect to canine breeds, sizes, and age. Despite advancements in our understanding, defining normative BAPs remains a challenge, leading to a lack of standardized data for canines.

## 6. Major Causes of Elevated TBA or Paired TBA

TBA or paired TBA is a useful liver function test. The TBA concentrations increase in the circulation when pathology alters the enterohepatic circulation. Decreased liver function, portosystemic vascular anomalies, and cholestasis are the most common reasons for its elevation [171,176].

### 6.1. Intrahepatic Diseases

When liver function is compromised, it decreases the removal of BAs from the portal circulation. Dysregulated BA metabolism affects lipid metabolism, immune environment, and intestinal bacteria, leading to inflammation, fibrosis, and hepatic steatosis. Gut microbes play a crucial role in regulating fat accumulation in hepatocytes, and changes in intestinal bacteria composition may affect immune function and inflammation [10,54,60,61]. The involvement of BAs and their receptors in liver repair and regeneration is critical. BA-induced proliferative and adaptive responses may play a role in both the biliary homeostasis maintenance and the regenerative response. During regeneration, BA receptors may collaborate to prevent toxic BA overload and restore a functional liver mass to its maximum capacity [185].

Yu et al. suggested that enhancing BA metabolism could be a promising therapeutic approach to prevent the onset and progression of non-alcoholic fatty liver disease (NAFLD) in human [60]. Also, Chow et al. published a report on NAFLD, which was characterized by the fat accumulation in the liver (macrosteatosis). This condition can evolve into a more severe version called nonalcoholic steatohepatitis, defined by inflammation and fibrosis. Steatohepatitis may occur as a result of an imbalance in BA pool. Presently, pharmacological therapy focuses on these routes [186].

The fecal BAPs of 28 adults with biopsy-confirmed NAFLD (non-alcoholic fatty liver: NAFL or non-alcoholic steatohepatitis: NASH), and healthy control subjects (HC) (*n* = 25) were analyzed in an investigation. In comparison to HC, patients with NAFLD had elevated concentrations of total fecal BA, CA, and CDCA and higher BA synthesis, according to the findings, similarly to previous publications on the serum/plasma BA content in individuals with NAFLD. In NASH, the proportion of primary to secondary BAs was greater than in the control group; however, there was no significant difference in the ratio of conjugated to unconjugated BAs between the groups. According to the study’s hypothesis, patients with NAFLD have distinct BA homeostasis, associated with dysbiosis, which renders them more susceptible to liver impairment. To this date, research of Mouzaki et al. offers the most exhaustive examination of BA homeostasis in patients diagnosed with NAFLD [111].

Sugita et al. used LC-MS/MS to measure the serum concentrations of 16 distinct BAs and compared them across various liver disorders. Patients with hepatitis, patients with alcoholic liver disease, patients with biliary tract disorders, NAFLD patients, and other liver patients were among them. As per the results, the group of liver patients with alcoholism had much higher levels of UDCA and glycoursodeoxycholic acid (GUDCA), while the group with biliary tract illness had significantly lower levels of both DCA and UDCA than the group with viral hepatitis [126].

Staley et al. performed an analysis regarding hepatic injury induced by BAs. The research group clarified that BAs could damage the integrity of cell membranes, having toxic consequences, which can make the liver more susceptible [17]. The lipophilicity of individual BAs dictates their ability to dissolve in fats, and the movement of conjugated BAs through transporters may impact their profile of causing cell death in drug-induced liver injury (DILI). BAs play a crucial role in the development of DILI that can result in severe and irreversible liver failure. The research demonstrated notable increases in BA levels in the blood and urine of rats exposed to several hepatotoxins, as well as alterations in metabolites related to energy, urea, and BA metabolism in the urine and blood. Changes in BA profiles may derive from hepatotoxicity or the functional blockage of BA transporters. These variations in BA composition in the circulation can occur independently of liver cell injury [59].

According to Slopianka et al., BAs (particularly CA, CDCA, DCA, muricholic acid (b), UDCA in plasma) may serve as further markers for diagnosing DILI in plasma, potentially even distinguishing between damage to hepatocytes and the biliary system [97]. Qin et al. also specifically studied rats with DILI. Although it was determined similarly that BAs play a critical role in pre-diagnosing DILI, their utility as biomarkers remained uncertain. Bile acids found in urine are more appropriate as biomarkers for identifying cholestatic liver injury than those in serum, as indicated by a research group’s findings [90].

According to the research by Rena-Ramos et al., dogs with cirrhosis and congenital circulatory abnormalities had the highest average resting serum bile acids (SBAs). Chronic hepatitis and congenital circulatory abnormalities were associated with the highest median PP concentrations [171]. The study included 341 dogs with various liver diseases. Liver biopsies were collected to evaluate the accuracy of resting and PP SBAs in detecting a variety of liver problems in dogs. The median resting SBAs were highest in dogs with cirrhosis and congenital circulatory abnormalities, while the conditions with the highest median PP levels were chronic hepatitis and circulatory anomalies. The resting SBA levels in dogs with liver illness across all groups ranged from less than 10 to more than 90 μmol/L [171].

Using ultra-performance liquid chromatography in tandem with mass spectrometry (UPLC-MS/MS), Zhang et al. assessed the plasma concentrations of 15 BAs in a total of 329 human subjects, including those with benign biliary diseases (BBD), CCA, gallbladder cancer (GC), hepatocellular carcinoma (HCC), and healthy individuals. Comparing HC and BBD groups to CCA patients, the majority of unconjugated BAs, such as CA, CDCA, UDCA, and LCA, were significantly lower in patients with CCA. On the other hand, from HC to BBD to CCA, there was a progressive stepwise rise in plasma-conjugated primary BAs, such as TCA, TCDCA, GCA, and glycochenodeoxycholic acid (GCDCA). Furthermore, compared to other groups, the absolute contents of total tauro-conjugated SBA and GUDCA were likewise significantly higher in CCA. Comparing the BBD groups to the malign groups, the multiple alterations of conjugated serum BAs in CCA were significantly smaller than those of conjugated primary BAs. Due to a decrease in total unconjugated BAs and an increase in conjugated BAs, the ratio of conjugated to unconjugated BAs was often significantly higher in CCA patients compared to the other study groups. Furthermore, CCA had considerably greater total primary BAs, but there was no difference in serum TBA across the three study groups. As a result, it was found that CCA patients had a lower SBA-to-primary-BA ratio. To sum up, distinct variations in the plasma concentration of BAs could function as diagnostic indicators to differentiate CCA from BBD and HC [141].

In a case report, the BAP of a dog with cholangiocarcinoma (CCA) and a healthy dog was compared by Thompson et al. When comparing the dog with neoplastic disorder to the healthy one, the former one’s total serum BA content was over a hundred times greater. TCA accounted for 88.8% of this total. Cholestasis was characterized by an elevated level and a predominance of TCA in the BAP, and a tumor-induced obstruction of the biliary tract was the cause of this condition. Furthermore, in this case, the BAP was identical to that published for rats and humans with cholestatic conditions [128].

In beagle dogs with cholestasis brought on by rifampicin, BAs were investigated. UPCL-MS/MS technology based on enzyme digestion was used to determine BAs. For nine days, the subjects were given 300 mg of rifampicin twice a day. Their blood was drawn fifteen minutes before the medication administration, and daily urine and intestinal sludge samples were taken between 0 and 24 h of treatment. Moreover, 27 BAs, including primary, secondary, and tertiary BAs, were examined. Following rifampicin treatments, daily urination and defecation revealed a rise in BA diversity in serum or urine but a decrease in BA diversity in feces, suggesting that rifampicin therapies changed the intestinal microbe composition of BAs metabolism [115].

### 6.2. Portosystemic Vascular Anomalies (PSVAs)

The portosystemic vascular abnormalities bypass the hepatic sinusoids and directly connect the portal venous system, containing contaminated splanchnic blood, with the systemic circulation. The above impairment is classified as congenital portosystemic shunts (CPSSs) or acquired PSSs (APSSs) [175]. Improper portal flow to the liver can impact liver function tests such as fasting ammonia (FA) or ammonia tolerance test (ATT), serum fasting BAs, and paired TBA concentrations [175,176]. A comparative study demonstrated that ceruletide-stimulated SBA testing is a viable substitute for postprandial SBA stimulation when it comes to diagnosing hepatic dysfunction, in particular, in canines that are concurrently experiencing upper respiratory disease and PSS [13].

Especially when evaluated postprandially, the serum TBA concentration is a dependable marker for PSS diagnosis with extraordinarily high sensitivity and specificity [187]. However, it is important to consider further aspects. Straten et al. demonstrated that in dogs showing symptoms, an elevation in both fasting TBA and FA are adequate for the diagnosis of PSS and may hold similar significance. Compared to FA concentrations, ATT and fasting TBA exhibit greater sensitivity in identifying PSS; however, they do not possess the same level of specificity as increasing FA concentrations. Combining fasting TBA with FA or ATT and fasting TBA in serial testing provides the benefit of enhancing specificity. The findings indicate that measuring fasting TBA concentrations is the most effective test for detecting CPSS in a healthy population [175,177].

Anglin et al. conducted an investigation involving 72 dogs diagnosed with PSS (cases collected between 2000 and 2008). The mean fasting BA values were as follows: 101.2 µmol/L at the initial presentation, 111.1 µmol/L at the first recheck examination, and 127.9 µmol/L at the second or final recheck examination (with a reference range of 0–10 µmol/L). The PP BA values were as follows: 145.6, 258.9 and 285.2 (with 0–30 µmol/L serving as the reference range) [188].

The majority of dogs with PSS had elevated concentrations of serum TBA according to a study involving 42 canines, where mean fasting values and mean PP values were 86.5 μmol/L and 165.5 μmol/L, respectively. This was the result of a median fasting period preceding the surgery [189].

Investigating the application of SBA as a prognostic indicator for congenital portosystemic shunt (PSS) in Maltese dogs, one study discovered that preoperative SBA concentrations were considerably greater (192 μmol/L) in this breed than those observed in other breeds (137 μmol/L) with portocaval shunt. However, postoperative SBA levels in Maltese and other dog breeds decreased following surgery. Regarding postoperative SBA levels, no significant difference was found between Maltese and other dog breeds. Measuring pre- and post-operative SBA levels may be a beneficial method for determining the prognosis of PSS in Maltese dogs, according to the study of Kim et al. [190].

The BA panel in dogs or other animals has not been examined for PSVA.

### 6.3. Cholestasis

A disrupted bile flow results in cholestasis, marked by increased levels of BA in the liver and bloodstream, leading to damage to hepatocytes and the bile epithelium [17]. Cholestasis may arise from either extrahepatic or intrahepatic causes. Li et al. concluded that despite extensive research into the etiology of cholestasis in humans, the precise molecular pathways, by which BAs induce liver damage, continue to be a subject of debate. As the research team described, in cholestatic liver injury, the ability to excrete BA is compromised due to either the direct inhibition or genetic defects of canalicular BA transporters in hepatocytes or due to a mechanical or immune-mediated blockage of the biliary ducts. Regardless of the underlying reason, there is an elevation in BA levels in both the liver and the bloodstream, which causes damage to hepatic cells and an increase in the growth of bile ducts. Untreated cholestatic liver damage frequently leads to the progression of liver fibrosis, cirrhosis, and ultimately liver failure. Li et al. conducted a comprehensive analysis of how BAs stimulate the formation of molecular agents that attract neutrophils, as well as the involvement of the inflammatory response in this pathogenic mechanism. These advancements indicate several new areas, where medications could potentially be considered useful treatments for cholestatic liver damage [79].

Pena-Ramos et al. suggested that the absence of an increase in SBAs in dogs could be caused possibly by bile being transported from the biliary tract into the duodenum ineffectively or slowly as a result of biliary epithelial inflammation. Due to the disrupted bile flow into the gut, BA absorption cannot occur properly. Consequently, two hours postprandially, the BA levels in the site of portal blood flow may not rise noticeably, which would restrict the rise in PP SBA concentrations in the systemic circulation in comparison with preprandial values [171].

Being consistent with research on humans, Jiang and Schnabl summarized that alterations in microbial composition were observed in an animal model of cholestasis (bile duct ligation) as early as three days following the procedure. In comparison to wild-type mice, the Mdr2/mouse, serving as an alternative mouse model of human primary sclerosing cholangitis (PSC), demonstrated an augmented translocation of bacteria and a modified composition of the intestinal microbiota. Significant intestinal dysbiosis and liver toxicity were induced when microbiota was transferred from Mdr2/mice to wild-type mice. This indicates a potential association between the gut microbiota and the progression of disease in Mdr2/mice. In summary, mounting evidence from clinical and preclinical investigations further substantiates the correlation between the microbiota and the cholestatic liver diseases. In a recent research study, alterations in the gastrointestinal microbiota were identified as a potential therapeutic target and biomarker for primary biliary cholangitis being a chronic, progressive consequence of cholestasis [61].

Increased damage to cholangiocytes may result from a protracted membrane exposure caused by the accumulation of hydrophobic BAs during cholestasis [17,51]. Bile salts accumulate and penetrate cholangiocytes extensively at low pH levels, in which the elevation in biliary acidity is a factor in the accelerated rates of apoptosis. The potential therapeutic applicability of UDCA in the prevention of cholestasis and the treatment of cholangiopathies is supported by its cytoprotective mechanisms. The activation of a BA nuclear receptor decreased systemic inflammation and improved cholangiocyte protection in mice but did not ameliorate primary sclerosing cholangitis, according to one study. Although it was observed to exacerbate pruritus induced by cholestasis, the protective properties of the nuclear receptor prompted research into its potential as a target for immune-mediated cholangiopathies [51].

Washizu et al. observed notable physiologic variations in the primary BAs, patterns of conjugation, and concentrations of various BAs in different species (dog, cow, horse, and human) [129].

Some years later, a biliary obstruction was induced in dogs via the surgical ligation of common bile ducts for the purpose of observing changes in the composition of SBAs. The results of the research suggest that the glycine/taurine ratio is not a practical indicator for detecting hepatopathy in canines. However, the CA:CDCA ratio may have a potential to fulfil this function [134].

The liver has a vital function in the systemic response to sepsis, and infectious bacteria can cause cholestasis and hyperbilirubinemia, even in the absence of direct hepatic invasion by the pathogen. Cholestasis occurs early in the sepsis process, and the SBA values were identified as an early indicator of short-term survival. Nevertheless, there is little evidence of the evaluation of SBAs in septic patients. Baptista et al. evaluated the significance of preprandial BA concentrations in a group of dogs diagnosed with sepsis. The hypothesis was that dogs with sepsis have elevated levels of resting SBAs. The data were retrospectively gathered over a 12-year period for analyzing data from 26 dogs, and they were contrasted in view of the SBAs of orthopedic patients with sepsis and healthy subjects. As a result, the septic group’s SBA levels were considerably higher than those of the non-septic groups in a positive correlation with the bilirubin levels. Nonetheless, there was no discernible change in the SBA level between the orthopedic and septic groups. However, the research had several limitations, including a large number of cases with incomplete data, a small sample size, and the use of single BA measurements [191].

A milder form of intrahepatic cholestasis is observed when circumstances such as hypoxia arising from centrolobular necrosis or enlarged hepatocytes due to hepatic glycogen, fat, or copper accumulation are present. Consequently, measurable BA values will increase albeit not to the same extent as major causes [192,193,194].

The intrahepatic cholestasis of pregnancy is a hepatic disorder characterized by elevated levels of sulfated BAs in the urine. One of the research projects developed a method to detect and measure sulfated BAs in urine by means of successfully isolating and quantifying 16 different forms of SBAs over a span of 18 min. The research revealed a significant increase in the overall levels of urinary sulfated BAs in individuals diagnosed with intrahepatic cholestasis, especially in relation to sulfated glycine-amidated and taurine-amidated BAs. The research proposes that urine sulfated BAs may serve as an alternative to SBAs for non-invasive sample collections in the diagnosis of the intrahepatic cholestasis of pregnancy [146].

Humbert et al. examined BA secretion and circulation in patients with exocrine pancreatic insufficiency (EPI) and chronic pancreatitis (CP). The investigation was conducted by concurrently quantifying PP concentrations of specific BAs in the contents of the duodenum and blood plasma. CP is a medical disorder characterized by a decrease in the body’s BAs, which leads to a diminished ability to absorb lipids. The levels and profiles of BAs in patients with exocrine pancreatic insufficiency did not undergo a comprehensive investigation. Humbert et al. revealed a significant decrease in the total BA levels in the duodenal contents of CP patients compared to those of healthy individuals. The study additionally found that patients with CP had decreased amounts of both glyco- and tauro-conjugated as well as unconjugated BA compared to the healthy group. Also, a contemporaneously elevated primary-to-secondary-BA ratio was experienced, which indicates an impaired transformation by the gut microbiota. The results indicated a significant 5-fold drop in the concentration of BA in the duodenal contents of patients with CP compared to healthy individuals. The study also discovered that patients with CP exhibited a significant rise in plasma BA concentration 30 min postprandially, which aligns with a diversion of BA circulation. These findings could provide an explanation for the 5-fold reduction in the BA pool observed in patients with CP compared to healthy individuals [102].

The physiological relationship between the composition of the intestinal microbiota and the metabolism of BAs suggests that changes in BA homeostasis may accompany dysbiosis, in this manner playing a role in the metabolic dysregulation observed in diabetes mellitus (DM). Alterations in both primary and secondary fecal unconjugated bile acids (fUBAs) were noted by Jergens et al. in dogs. These changes were characterized by elevated amounts of CA and the total primary fUBAs. On the other hand, LCA levels were lower, both as a proportion of all fUBAs and as an absolute number, when measured in dogs with diabetes. DM occurring naturally in dogs is insulin-dependent and may be accompanied by dysbiosis and an altered BA metabolism. Consequently, they may provide a clinically applicable model for analyzing human disease [154].

BAs perform endocrine functions, and they are implicated in glucose homeostasis. The interaction with the gut microbiome was demonstrated to be altered in humans and rodents with type 1 and type 2 diabetes. This connection further complicates the metabolic function of BAs [195]. By analyzing the serum metabolomic profiles of diabetic canines, O’Kell et al. identified parallels to type 1 diabetes in humans in their pilot study. They revealed that primary and secondary BAs (TCDCA, TDCA, and TUDCA), among other metabolites, were substantially lower in diabetic dogs compared to healthy dogs. Furthermore, disruptions in a variety of BAs were recognized in humans. The investigation displayed that metabolomics provides the potential of distinguishing between diabetic and non-diabetic dogs [78].

The concentrations of total taurine-conjugated BAs were found to be intermediate in impaired individuals and higher in type 2 diabetes according to Wewalka et al. upon comparing the SBA composition of individuals with normal glucose tolerance, type 2 diabetes, and impaired glucose tolerance. Positive associations with fasting and postload glucose levels were observed, whereas insulin resistance did not correlate with this association [196].

### 6.4. Other Causes of Elevated TBA Level

Microbiomes are critically involved in BA metabolism. Dysbiosis is characterized by the altered deconjugation of BA, particularly when *Clostridium hiranonis* populations are decreased. As a consequence, the efficiency of BA removal from portal circulation is compromised, leading to an accumulation of circulating total BAs. The research by Melgarejo et al. aimed to determine the diagnostic value of unconjugated BA concentrations in the serum of canines with bacterial proliferation. Fasting sera were collected from 23 dogs in all, from which 10 had dysbiosis (previously SIBO) confirmed by culture, 8 were diagnosed indirectly, and 5 were HC. Unconjugated BA concentrations were significantly elevated in dogs with SIBO and dogs with indirectly diagnosed SIBO. The unconjugated BA CA predominated in the serum of dogs afflicted with SIBO. The study emphasizes the importance of detecting and managing SIBO in dogs to prevent transmission and facilitate treatment. The high capacity of liver to conjugate BAs to amino acids taurine and glycine makes conjugated primary BAs constitute nearly all of normal hepatic bile. Serum unconjugated BA measurements provide a noninvasive, specific, and sensitive diagnostic tool for SIBO in canines. The effects of dietary taurine and glycine on canine BA conjugation were investigated. The results of the study in question, for the first time, evidenced that dogs with bacteriologically proven SIBO have a higher rate of intestinal BA deconjugation than normal healthy pet dogs [123]. However, the diagnostic value of these results was not verified by other researchers [123,197].

Comito et al. used high-performance liquid chromatography in tandem with mass spectrometry (HPLC-MS/MS) to conduct a thorough study of 31 compounds from among fecal BAs and related microbiota metabolites. The diseased participants had noticeably greater levels of CA in their stool. Additionally, pathological individuals had higher CDCA than the control participants. Conversely, the control subjects had greater DCA and LCA values for fecal sludge. Significantly increased β-muricholic acid, decreased 3,12-dioxo-DCA, and 3-oxo-LCA levels were found in relation to the other microbiota metabolites. For other BAs, no discernible variations were discovered. The number of distinct BA groups (primary, secondary, and oxo-BA), as well as the overall number of non-oxo-BAs and BAs were computed in order to comprehend the impact of CIE on the BA pool. The intestinal sludge of dogs with CIE had a notably greater primary BA concentration than for the control subjects [198].

Corticosteroids regulate fecal BAs in canines with CIE, according to a study by Guard et al. The patients had an elevated DI at baseline, which remained unchanged over time. In comparison to healthy dogs, secondary fUBA was reduced in dogs with CIE; nevertheless, the percentage of secondary fecal UBA increased after two to three months of treatment. Guard et al. observed BA malabsorption either as a primary condition or in combination with chronic inflammatory bowel disease (IBD) [83].

Untargeted metabolomics is an emerging discipline that investigates metabolic pathways in fecal samples. In a comparative analysis of metronidazole and fecal microbial transplantation (FMT) for canines with acute diarrhea, notable distinctions were observed in terms of fecal consistency, microbiota composition, and metabolome profiles. Fecal consistency improved with both FMT and the antibiotic agent, but FMT demonstrated superior consistency. In terms of bacterial taxa, both groups differed considerably. Although FMT treatment resulted in reduced percentages of primary BAs, metronidazole was unable to restore these profiles to normal. The hypothesis of the study, i.e., FMT could restore the composition of BAs in the feces, was confirmed. Thus, metronidazole failed to restore fecal metabolome, whereas FMT did. The investigation concluded that canines with acute diarrhea exhibited a normalization of microbiota and an increase in microbial diversity. On the other hand, fecal metabolomes continued to exhibit dysbiosis even after a period of 28 days [149].

Another study also confirmed that FMT could restore the initial balance of the BAP after antibiotic treatment-induced damage [17]. Belchik et al. also confirmed the observation that the administration of antibiotics led to BA dysmetabolism and intestinal dysbiosis in dogs [184].

Tylosin significantly altered the BAP and fecal microbiome of healthy canines in the investigation conducted by Marclay et al., which resulted in a decrease in secondary BA concentrations on day 7. However, on day 14, the majority of parameters returned to the baseline. BA dysmetabolism and intestinal dysbiosis were observed in canines as a result of this macrolide antibiotic, but the impact of FMT on the recovery of the microbiome and metabolome remained uncertain. After administering the drug, the values returned to the baseline [157]. In contrast, Manchester et al. observed that fecal dysbiosis and corresponding shifts in fecal unconjugated BAs did not resolve uniformly in healthy canines after tylosin cessation [156,157].

By raising secondary BA concentrations, *C. scindens* was shown, in a study by Marclay et al., to assist a mouse model fight infection from *C. difficile*. This suggests that manipulating bacteria metabolizing BAs may be a viable therapy for individuals susceptible to infections of this nature. In the investigation, antibiotic dosing influenced all particular BA concentrations and ratios but did not affect total BA concentrations [157].

Primary BAs significantly increased, while secondary BAs considerably decreased under the treatment of antibiotics. Rather than indicating alterations in re-absorption, the lack of influence on total BA points to a reduced transformation of primary to secondary BA by GI microorganisms [157].

Secondary BA concentrations were lower in dogs with GI disease compared to the healthy ones. In order to ascertain the clinical relevance of lactate and BA quantification in canine feces, additional research is required. While these metabolites may not serve as reliable indicators of dysbiosis on their own, they become promising targets for further investigations in the role of microbiota in health and disease when interpreted in conjunction with bacterial abundances [65].

The available studies, in which profiles of BAs were measured, are summarized by the conditions of dogs in the following table (Table 3).

In summary, the bibliographical evidence regarding the association between BAs and the disorders in question in canines is very far from solid. Furthermore, a sufficiently diverse profile of BAs is absent from the available studies to determine whether a well-defined pattern exists in BAPs for a particular canine disease. In cases where abnormalities are suspected, clinicians would also require clarification regarding the determination of whether feces or blood should be collected for the purpose of measuring BAPs. There is a lack of research that has compared the therapeutic benefits of concurrently taking blood and feces samples for BAP determination in identical conditions; nevertheless, such an analysis could reveal whether the profiles derived from the two matrices demonstrate parallel patterns or a correlation of any kind. Such research could potentially yield valuable insights and thereby provide therapeutic alternatives.

## 7. Conclusions

The examination of bile acids provides essential knowledge in various fields of health science, including physiology, microbiology, internal medicine, and pharmacology. They undergo a complex transformation in the gut, affecting their structure and biochemical properties, acquiring several types of physiologic roles. Classified based on their impact on organisms, bile acids are categorized as primary or secondary, and they are further differentiated by their hydroxylation status (mono-, di-, or trihydroxy) and conjugation (taurine- or glycine-conjugated, or unconjugated). Perturbations in gut microbiota composition can disrupt bile acid metabolism, which may lead to various pathological conditions in human as well as in veterinary cases.

In clinical diagnostic laboratories, the enzymatic reaction-based photometric method remains a fundamental tool for measuring total bile acids, owing to its simplicity and cost-effectiveness. On the other hand, there is an increasing demand for more comprehensive bile acid panels, which are likely to be acquired by the LC-MS/MS technique. While triple quadrupole (QqQ) mass spectrometers currently dominate routine targeted analyses, the expanding scope of non-targeted metabolomic studies is expected to elevate the significance of time-of-flight (TOF) and ion trap (IT) instruments in bile acid analysis. This evolution reflects a growing demand for detailed and nuanced assessments in clinical practice.

Bile acid levels in dogs are typically assessed following a 12 h fasting period, with measurements taken 2 h after the consumption of canned food to obtain paired values for comparison. Various factors influence bile acid circulation in canines, including meal composition, intestinal transit times, motility, and the gut microbiome, and obviously, the composition of the bile acid pool varies among dog breeds, sizes, and ages.

Multiple matrices, such as blood, feces, urine, liver tissue, and gallbladder bile, can be used to determine canine bile acid profiles. While many studies focus on analyzing a single matrix, achieving a precise diagnostic outcome may necessitate the examination of multiple matrices to obtain a comprehensive understanding of the bile acid pool. Although blood serum or plasma is often preferred for analysis, the choice of matrix depends on technical expertise and laboratory objectives. Despite feces being the second most commonly analyzed matrix in the literature, its interpretation remains challenging due to significant individual and breed-specific variations, as well as technical sampling difficulties (since sample collection conditions cannot be adapted from laboratory rodents or humans to dogs). Nonetheless, comparing bile acid levels between feces and biological fluids could provide valuable insights into physiological and pathological changes in the enterohepatic circulation of bile acids. Altogether, there is a significant shortage of research that thoroughly examines bile acid profiles in various matrices, especially taking into account breed- and age-specific differences and clinical situations, even in healthy dogs.

In clinical trials, the combined measurement of various bile acids is gaining increased attention; however, a consensus among experts regarding the specific matrix to examine remains precarious. While certain bile acids are measured individually and TBA is quantified, the range of compounds measured is frequently insufficiently defined and/or insufficiently broad to derive diagnostic and prognostic significance. Furthermore, the existence of a distinct pattern for each disease remains unknown. The bibliographical material regarding the association between BAs and a specific disorder in dogs is far from established. It is also noteworthy to highlight that based on the information presented in this study, the success of using the Beagle breed for experimental purposes to examine the bile acid profile is also highly doubtful.

Our review endeavored to navigate the ongoing debates concerning the diverse beneficial and detrimental properties of different bile acids while also elucidating the physiological effects associated with their various groups. Addressing a significant gap in the literature, our review shed light on species-specific variations observed in the measurement of bile acid parameters, with a particular emphasis on canines. We underscored the importance of utilizing appropriate matrices alongside the potential diagnostic implications derived from them, highlighting disparities between human and canine results and sampling methodologies. This differentiation is crucial given the distinct digestive system characteristics in dogs, which encompass disparate processes and microbiological composition.

Further investigations into alterations of the overall bile acid profile in canine pathological conditions would significantly contribute to a more comprehensive understanding of various pathomechanisms, interrelationships among companion animal illnesses, and the correlation between the gut microbiome and the extensive systemic effects of bile acids on the entire body. Additionally, a thorough exploration of the interactions between bile acids and the gastrointestinal microbiome holds promise for novel therapeutic avenues. Therefore, pursuing these research objectives presents an intriguing prospect from both a scientific and therapeutic standpoint.

## Figures and Tables

**Figure 2 metabolites-14-00178-f002:**
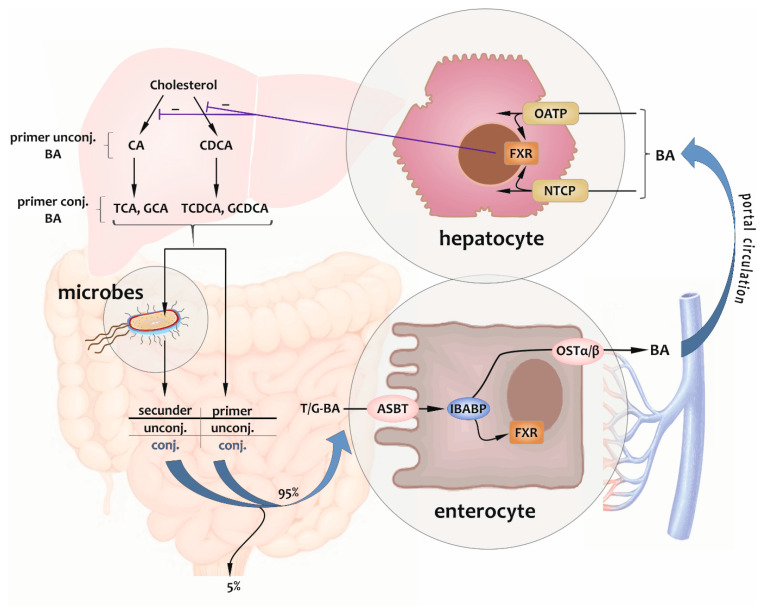
Enterohepatic circulation of bile acids (BAs). BAs are synthesized in the hepatocytes and then, after tauro- or glycoconjugation, secreted into the lumen of the small intestine via bile following their production by the liver. In the distal segment of the small intestine, enterocytes recover the majority of conjugated BAs through an active mechanism involving sodium-dependent BA transporter (ASBT) embedded in the apical membrane. Bile acids that are not absorbed (5% of the total amount) then travel to the colon, where they are modified further by the gut microbes, while 95% of the BAs stay in the cycle. In ileal enterocytes, ileal BA-binding protein (IBABP) serves as the primary intracellular transporter for reabsorbed conjugated BAs. The heterodimeric basolateral membrane organic solute transporter OSTα/OSTβ facilitates the elimination of BAs from ileal enterocytes, allowing them to be transported back to the liver through the portal circulation. Bile acids are reabsorbed into the hepatocytes, assisted by sodium taurocholate co-transporting polypeptide (NTCP) and organic anion-transporting polypeptides (OATPs), which are located in the basolateral (sinusoidal) membrane of the liver in direct contact with the portal blood. The occurrence of this uptake mechanism has negative feedback on the process of endogenous synthesis from cholesterol. Since the cycle is continuous, the liver also excretes these reabsorbed BAs into the bile [20,36,37,49].

**Figure 3 metabolites-14-00178-f003:**
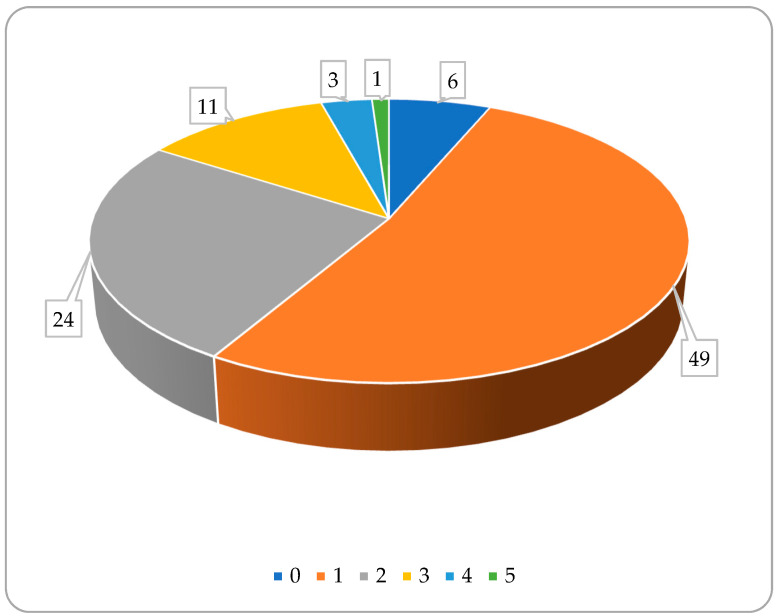
Number of matrices tested for BAs within a trial (*n* = 94) (number of relevant papers).

**Figure 4 metabolites-14-00178-f004:**
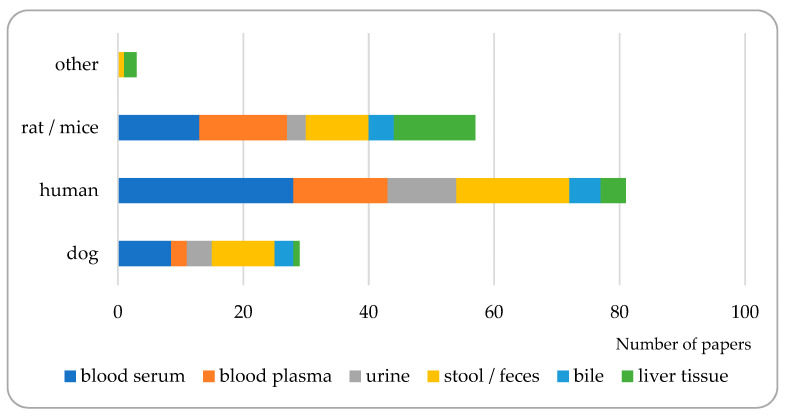
Matrices studied the most frequently for BA composition by species (in number of papers; *n* = 94).

**Figure 5 metabolites-14-00178-f005:**
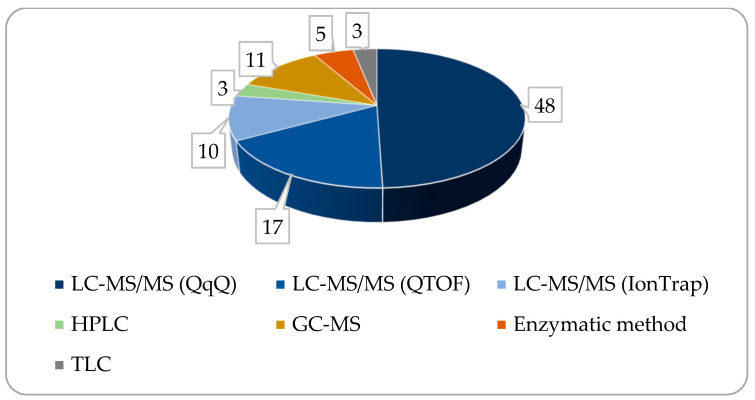
Ratio and number of measurement methods in the bibliographic sources processed (*n* = 94).

**Table 1 metabolites-14-00178-t001:** Molecular structure, common name, chemical name, and abbreviated names of most relevant primary and secondary conjugated and unconjugated bile acids in dogs [15,18,33,34].

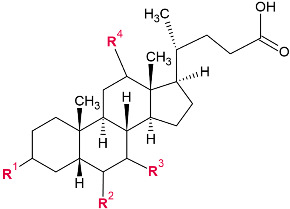			
Common Name	Chemical Name	Abbreviation	
cholic acid	5b-cholanic acid-3a,7a,12a-triol	CA	**PRIMARY BILE ACIDS**
taurocholic acid	5b-cholanic acid-3a,7a,12a-triol-N-(2-sulpho-ethyl)-amide	TCA	
glycocholic acid	5b-cholanic acid-3a,7a,12a-triol-N-(carboxymethyl)-amide	GCA	
chenodeoxycholic acid	5b-cholanicacid-3a,7a-diol	CDCA	
taurochenodeoxycholic acid	5b-cholanic acid-3a,7a-diol-N-(2-sulpho-ethyl)-amide	TCDCA	
glycochenodeoxycholic acid	5b-cholanic acid-3a,7a-diol-N-(carboxymethyl)-amide	GCDCA	
deoxycholic acid	5b-cholanic acid-3a,12a-diol	DCA	**SECONDARY BILE ACIDS**
taurodeoxycholic acid	5b-cholanic acid-3a,12a-diol-N-(2-sulphoethyl)-amide	TDCA	
glycodeoxycholic acid	5b-cholanic acid-3a,12a-diol-N-(carboxymethyl)-amide	GDCA	
lithocholic acid	5b-cholanic acid-3a-ol	LCA	
taurolithocholic acid	5b-cholanicacid-3a-ol-N-(2-sulphoethyl)-amide	TLCA	
glycolithocholic acid	5b-cholanic acid-3a-ol-N-(carboxymethyl)-amide	GLCA	
ursodeoxycholic acid	5b-cholanic acid-3a,7b-diol	UDCA	
tauroursodeoxycholic acid	5b-cholanic acid-3a,7b-diol-N-(2-sulphoethyl)-amide	TUDCA	
glycoursodeoxycholic acid	5b-cholanic acid-3a,7b-diol-N-(carboxymethyl)-amide	GUDCA	

**Table 3 metabolites-14-00178-t003:** Studies on bile acid profiles in canines.

Bibliographic Source	Illness(es)
[115]	rifampicin-induced cholestasis (DILI)
[112]	healthy, hepatic disorders, clinical signs of hepatic disorder
[133]	gallbladder mucocele formation
[134]	biliary obstruction produced by surgical ligation of the common bile duct
[123]	culture-proven SIBO, indirectly diagnosed SIBO
[128]	metastatic cholangiocarcinoma
[129]	none (healthy dogs)
[130]	none (healthy dogs)
[144]	none (healthy dogs)
[143]	none (healthy dogs)
[3]	none (healthy dogs)
[65]	gastrointestinal diseases: chronic enteropathy (CE) and exocrine pancreatic insufficiency (EPI)
[19]	none (healthy dogs)
[149]	acute diarrhea (AD)
[83]	steroid-responsive chronic inflammatory enteropathy (CE)
[52]	none (healthy dogs)
[154]	insulin-dependent diabetes mellitus
[156]	none (healthy dogs)
[157]	none (healthy dogs)
[158]	none (healthy dogs)
[99]	none (healthy dogs)

## Data Availability

Not applicable.

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
