# Peer review of "Determination of Bile Acids in Canine Biological Samples: Diagnostic Significance"

_metabolites, 2024, doi:10.3390/metabo14040178_

Round 1

Reviewer 1 Report

Comments and Suggestions for Authors

The manuscript by Nemeth et al is a review of bile acids, mainly in veterinary animals. The review has a clear structure and is written in a way that can be understood by a non-expert in bile acids. The only major criticism is that the text is long.

Comments:
The words "de novo" and "et al." were not italicised.
The quality of the structure shown in Table 1 should be improved.
Figure 3: Please explain the colour coding and the numbers in the figure legend.
Line 487: The abbreviation for Ultra High Performance Liquid Chromatography is UHPLC. UPLC stands for Ultra Performance Liquid Chromatography. This is quite confusing, so some researchers use UHPLC for ultra-high pressure liquid chromatography because Waters patented the same ultra-performance liquid chromatography (UPLC).
Line 500: The term "mother ion" is no longer used. Recently the term precursor ion or molecular ion is used.
Line 570-511: The sentence is written in such a complicated way that it is very difficult to understand. Please reformulate. You can split the sentence into two parts. In the first sentence, you can write that reversed-phase liquid chromatography with C18 columns is most commonly used. In the second sentence, you could describe the mobile phases used for gradient elution.
Line 517: TQ is also a specific instrument. TQ, as well as TOF and IT, are analysers of the mass spectrometer. The difference between TQ and TOF/IT is that the latter separates gaseous ions with high resolution. What is specific to untargeted analysis is data mining, as specific pre-processing procedures and sophisticated statistical methods are required to perform metabolite profiling of bile acids.
Line 622: a "(" is missing.
Line 690: The style of the reference differs from the others.

Author Response

Reviewer A:

The manuscript by Nemeth et al is a review of bile acids, mainly in veterinary animals. The review has a clear structure and is written in a way that can be understood by a non-expert in bile acids. The only major criticism is that the text is long.

-> The authors removed some part of the text in order to make it somewhat shorter.

Comments:

The words "de novo" and "et al." were not italicised.

-> It has been corrected.

The quality of the structure shown in Table 1 should be improved.

-> The structure has been modified.

Figure 3: Please explain the colour coding and the numbers in the figure legend.

-> The legend has been modified.

Line 487: The abbreviation for Ultra High Performance Liquid Chromatography is UHPLC. UPLC stands for Ultra Performance Liquid Chromatography. This is quite confusing, so some researchers use UHPLC for ultra-high pressure liquid chromatography because Waters patented the same ultraperformance liquid chromatography (UPLC).

-> The abbreviation has been modified.

Line 500: The term "mother ion" is no longer used. Recently the term precursor ion or molecular ion is used.

-> It has been corrected.

Line 570-511: The sentence is written in such a complicated way that it is very difficult to understand. Please reformulate. You can split the sentence into two parts. In the first sentence, you can write that reversed-phase liquid chromatography with C18 columns is most commonly used. In the second sentence, you could describe the mobile phases used for gradient elution.

-> The sentence has been divided into three shorter parts.

Line 517: TQ is also a specific instrument. TQ, as well as TOF and IT, are analysers of the mass spectrometer. The difference between TQ and TOF/IT is that the latter separates gaseous ions with high resolution. What is specific to untargeted analysis is data mining, as specific pre-processing procedures and sophisticated statistical methods are required to perform metabolite profiling of bile acids.

-> The section has been modified accordingly.

Line 622: a "(" is missing.

-> The citation has been corrected.

Line 690: The style of the reference differs from the others.

-> The citation has been corrected.

Reviewer 2 Report

Comments and Suggestions for Authors

Overall, the paper is well-written and does a good job of going into all of the relevant factors impacting the levels of bile acids.  Based on the title of the review, a longer discussion of the analytical methodologies was expected.  The analytical section is quite concise and does not go into the level of detail that the rest of the review appears to dive into.  In fact, the section appears to be an after-thought.  Either the title should be rewritten to more accurately reflect this or the section should be strengthened.

Figure 4. It's fairly clear but it would be worth adding an axis title to the y-axis to clarify that it's count and not percentage.

Line 481 and below.  Specific detail has gone into the reasons for using different LC and MS methods, but no mention of the need for tandem MS methods for overlapping masses abundant in the bile acids that UHPLC cannot separate as one of the biggest challenges in quantifying this class of metabolites.

Author Response

Reviewer B:

Overall, the paper is well-written and does a good job of going into all of the relevant factors impacting the levels of bile acids.

Based on the title of the review, a longer discussion of the analytical methodologies was expected. The analytical section is quite concise and does not go into the level of detail that the rest of the review appears to dive into. In fact, the section appears to be an after-thought. Either the title should be rewritten to more accurately reflect this or the section should be strengthened.

-> The title of manuscript has been modified. Amending the analytical section was considered and dismissed since the text is already rather long.

Figure 4. It's fairly clear but it would be worth adding an axis title to the y-axis to clarify that it's count and not percentage.

-> Axis title has been added.

Line 481 and below. Specific detail has gone into the reasons for using different LC and MS methods, but no mention of the need for tandem MS methods for overlapping masses abundant in the bile acids that UHPLC cannot separate as one of the biggest challenges in quantifying this class of metabolites.

-> The section has been modified accordingly.